# Genome-Wide Identification and Expression Analysis of m^6^A Methyltransferase Family in *Przewalskia tangutica* Maxim.

**DOI:** 10.3390/ijms26083593

**Published:** 2025-04-11

**Authors:** Xing Ye, Xingqiang Hu, Kun Zhen, Jing Meng, Heyan Du, Xueye Cao, Dangwei Zhou

**Affiliations:** 1The College of Pharmacy, Qinghai Minzu University, Xining 810007, China; xingye202203@163.com (X.Y.); 17736051087@163.com (X.H.); m13511431841@163.com (K.Z.); 15508053103@163.com (J.M.); 15885044089@163.com (H.D.); 15997072052@139.com (X.C.); 2Key Laboratory for Tibet Plateau Phytochemistry of Qinghai Province, Xining 810007, China; 3Key Laboratory of Adaptation and Evolution of Plateau Biota (AEPB), Northwest Institute of Plateau Biology, Chinese Academy of Sciences, Xining 810008, China

**Keywords:** m^6^A methyltransferase, *Przewalskia tangutica* Maxim., phylogenetics, expression analysis

## Abstract

*N*^6^-methyladenosine (m^6^A) RNA modification plays important regulatory roles in plant development and adaptation to the environment. However, there has been no research regarding m^6^A RNA methyltransferases (MT-A70) in *Przewalskia tangutica* Maxim. Here, we performed a comprehensive analysis of the MT-A70 family in *Przewalskia tangutica* (*PtMTs*), including gene structures, phylogenetic relationships, conserved motifs, gene location, promoter analysis, GO enrichment analysis, and expression profiles. We identified seven *PtMT* genes. Phylogeny analysis indicated that the seven *PtMT* genes could be divided into three groups; two MTA genes, three MTB genes, and two MTC genes, and domains and motifs exhibited similar patterns within the same group. These *PtMT* genes were found to contain a large number of cis-acting elements associated with plant hormones, light response, and stress response, suggesting their widespread regulatory function. Furthermore, the expression profiling of different tissues was investigated using RNA-seq data, and the expression of seven genes was further validated by qPCR analysis. These results provided valuable information to further elucidate the function of m^6^A regulatory genes and their epigenetic regulatory mechanisms in *Przewalskia tangutica*.

## 1. Introduction

Over 160 RNA modifications have been recognized in eukaryotes, with prevalent modifications comprising *N*^6^-adenylated methylation (m^6^A), *N*^1^-adenylate methylation (m^1^A), and cytosine hydroxylation (m^5^C) [1,2,3]. Among these, m^6^A is the predominant internal modification in eukaryotic RNA, extensively present in mRNA, tRNA, miRNA, and long non-coding RNA [4]. As a dynamic and reversible modification, m^6^A is essential in post-transcriptional regulation and affects multiple facets of RNA metabolism, such as gene expression regulation, RNA editing, and the management of mRNA stability and degradation, rendering it a focal point of considerable research interest [5,6,7,8].

m^6^A is a highly conserved chemical modification that arises when the hydrogen at the N6 position of adenosine is substituted with a methyl group (CH3) [9]. This alteration was initially discovered in Novikoff liver carcinoma cells in 1974 [10]. Subsequently, m^6^A RNA methylation has been documented in multiple species, including mouse (*Mus musculus*) [11], *Drosophila melanogaster* [12], wheat (*Triticum aestivum*) [13], oats (*Avena sativa*) [14], and *Saccharomyces cerevisiae* [15]. The enzymes implicated in m^6^A modification are classified into three categories: “writers”, “erasers”, and “readers” [16,17]. The “writers” denote m^6^A methyltransferases that assemble into a complex to identify and methylate target mRNAs. This complex in animals consists of METTL3, METTL14, WTAP, KIAA1429, and VIRILIZER [18,19,20]. In plants, the METTL3 homolog MTA (adenosine methylase) and the METTL14 homolog MTB (methyltransferase B) function as essential subunits of the plant m^6^A methyltransferase complex. Both possess conserved MT-A70 structural domains, are situated in the nucleus, and form heterodimers [18,21].

MTA and MTB are intricately linked to m^6^A synthesis. Although the majority of m^6^A-related research has been performed in animals, especially humans [22,23], investigations in plants are still comparatively scarce. Nevertheless, advancements in detection methodologies have led to a gradual increase in plant-related research, particularly involving *Arabidopsis thaliana* and rice (*Oryza sativa*) [24]. A homology-based search identified 5 m^6^A writers, 13 readers, and 13 erasers in *Arabidopsis thaliana* [25,26]. Zhong et al. [27] employed two-dimensional thin-layer chromatography (TLC) to verify the existence of m^6^A modifications in MTA-deficient seeds. The m^6^A modification level was diminished by roughly 50% in the RNAi strain of Arabidopsis MTB relative to the wild type [21]. In rice, four genes homologous to Arabidopsis MTA were identified, with only the *MTA2* mutation significantly diminishing m^6^A levels, whereas the other three genes did not influence overall m^6^A levels [28]. Homologous proteins of MTA and MTB have recently been identified in strawberries (*Fragaria vesca*), where they also function as m^6^A methyltransferases [29]. These findings collectively demonstrate that m^6^A serves a crucial regulatory function in plant growth, development, and responses to abiotic stresses.

*Przewalskia tangutica* Maxim. (*P. tangutica*), a perennial species within the Solanaceae family, predominantly resides in high-altitude areas ranging from 3200 to 5000 m. This plant has been used as a medicinal remedy for centuries, with its roots, seeds, and entire structure employed for analgesic, antispasmodic, and anti-inflammatory purposes [30]. Previous study has shown that the roots of *P. tangutica* are abundant in scopolamine and anisodamine [31]. The comprehension of m^6^A methyltransferases in *P. tangutica* is still constrained. This study involved a comprehensive genome-wide characterization of *PtMT* genes using recently published genomic data [32]. We conducted an exhaustive analysis of their phylogenetic relationships, gene and protein structures, chromosomal organization, cis-regulatory elements, expression patterns, and physicochemical properties. This study seeks to create a foundational framework for investigating the functional roles of m^6^A regulatory pathway genes in plants like *P. tangutica*.

## 2. Results

### 2.1. Identifying the PtMT Genes

By screening the whole genome data of *P. tangutica*, seven *PtMT* genes were finally identified (Table 1), the length of the seven *PtMT* genes ranged from 867 to 3285 bp, with an average of 2324 bp. The amino acid lengths varied from 288 to 1094 aa, with an average number of amino acids of 773aa. The average molecular weight (Mw) is 86.63 kDa, the theoretical pI ranged from 5.12 to 7.1, and the seven *PtMT* genes are essentially acidic. The instability index is greater than 40 and the aliphatic index ranged from 48.39 to 83.26. The grand average of hydropathicity (GRAVY) analysis showed that PtMT proteins were hydrophilic proteins. Subcellular localization prediction displayed that four genes (57.1%) were located in the nucleus, two genes (*PtMTA1*, *PtMTA2*) (28.6%) were located in the chloroplast, and one gene (*PtMTC2*) (14.3%) was located in the cytoplasm.

### 2.2. Phylogenetic Relationship Analysis

We first identified m^6^A methyltransferases (MT-A70 like proteins) in *P. tangutica* and other species, then constructed a phylogenetic tree using these proteins (Figure 1). The results showed that the seven *PtMT* genes in *P. tangutica* could be categorized into three subfamilies, MTA, MTB, and MTC, and each subfamily contained at least two *PtMT* genes, of which the MTB subfamily was the largest, accounting for nearly half of the total. In terms of kinship, *P. tangutica* is closest to the *Anisodus acutangulus* and *Atropa belladonna PtMT* gene families.

### 2.3. Structural Features Analysis

To analyze the diversity and distribution patterns of conserved motifs, we applied the MEME online software (https://meme-suite.org/meme/, accessed on 15 February 2024) to analyze the amino acid sequences of seven MT-A70 proteins (Figure 2A). We identified ten different motifs, named Motif 1 to Motif 10 (Appendix A), respectively. These motifs were highly similar and had the same arrangement order within the same subfamily, indicating that all the proteins were highly conserved. However, there were certain differences among different subfamilies. Among them, Motif 5 was detected in all the proteins of the family, suggesting that Motif 5 was the most conserved motif in the MT-A70 protein sequences.

We conducted an analysis of the exon–intron structure of the seven *PtMT* genes. The results showed (Figure 2B) that the numbers of exons and introns in the three subfamilies were not consistent, while the numbers within the same subfamily were relatively conserved. In addition, most of the introns of the *PtMT* genes were longer than those in the corresponding *Arabidopsis thaliana*, which is consistent with the fact that *P. tangutica* has a larger genome. The numbers of exons of the seven *PtMT* genes ranged from 6 to 10. Among them, the *PtMTB1*, *PtMTB2*, and *PtMTB3* genes contained 6 exons, *PtMTA1* and *PtMTA2* contained 7 exons, *PtMTC1* contained 10 exons, and *PtMTC2* contained 8 exons. This indicates that the members of the *PtMT* genes family have differentiated during the process of evolution, and there have been phenomena of exon increase or deletion. The analysis of conserved domains showed that the *PtMT* genes family all contained the typical MT-A70 domain (Appendix A).

### 2.4. Chromosomal Location and GO Enrichment Analysis

To further understand the evolutionary relationships of the 7 *PtMT* genes, we determined the positions of these genes on the chromosomes. The results showed that these genes were distributed across five chromosomes (Figure 3A). Specifically, chrA01 and chrA12 each contained two genes, while chrA08, chrA19, and chr20 each contained one gene. Most of the *PtMT* genes were located at the distal ends of the chromosomes (*PtMTA1*, *PtMTB1*, *PtMTA2*, and *PtMTC1*), and very few were located in the central regions. The GO enrichment analysis revealed that the seven *PtMT* genes played roles in molecular functions, cellular components, and biological processes (Figure 3B and Appendix A).

### 2.5. Promoter Analysis of PtMT Genes

Cis-regulatory element analysis showed that a total of 147 cis-elements were identified in *PtMT* genes (Figure 4). When grouped according to the biological processes in which they participate, the cis-elements were classified into six categories (Appendix A). The largest group was hormone response related, including 72 (48.98%) elements, such as abscisic acid (ABA) response elements (ABRE), methyl jasmonate (MeJA) response elements (CGTCA-motif), gibberellin (GA) response elements (GARE-motif and TATC-box), and salicylic acid (SA) response elements (TCA-element). Among the plant-hormones-related elements, MeJA-responsive elements and ABA response elements were the largest two groups. The second largest group was light response related, comprising 43 (29.25%) elements, such as AE-box, Box 4, Gap-box, G-box, MRE, Sp1, and CAG-motif.

### 2.6. The 3D Protein Structure Analysis

The seven proteins encoded by the *PtMT* genes were subjected to secondary structure prediction using GOR IV, and the results are shown in Appendix A. The secondary structure analysis indicated that the proteins encoded by the *PtMT* genes were mainly composed of alpha-helices (Hh), extended strands (Ee), and random coils (Cc). Among them, random coils accounted for the largest proportion (50.68%~66.67%), followed by alpha-helices (19.13%~31.71%) and extended strands (14.67%~31.25%). The secondary structures of the screened proteins were approximately the same. The PtMTB3 protein had the highest proportion of random coils (66.67%) and the PtMTC2 protein had the lowest proportion of random coils (48.61%), it is speculated that these two proteins may have special functions. The three-dimensional structure prediction showed that the structures of PtMTA1 and PtMTA2 proteins were similar, and the structures of PtMTB1, PtMTB2, and PtMTB3 proteins were similar, indicating that they had the same functions (Figure 5).

### 2.7. Collinearity Analysis

The results of collinearity analysis of *PtMT* genes are shown in Figure 6. *PtMTA1* and *PtMTB1* genes were distributed on chr02, *PtMTB2*, and *PtMTB3* genes on chr12, *PtMTA2*, *PtMTC1*, and *PtMTC2* genes on chr08, chr19, and chr20, respectively. Three gene pairs existed in a collinearity relationship, *PtMTA1* and *PtMTA2*, *PtMTB1* and *PtMTB2*, and *PtMTC1* and *PtMTC2*. The collinearity of *PtMT* gene families was further illustrated using DotPlot (Appendix A). To better understand the evolutionary history of *PtMT* gene families, the *Ka/Ks* values of the three collinear gene pairs were calculated using the TBtools software (v2.096) (Appendix A), The results showed that the three collinear gene pairs belong to segmental duplication, and the *Ka/Ks* values of the three pairs were all less than 0.5, indicating that purifying selection has always existed during the evolutionary process of the *PtMT* gene family.

### 2.8. Expression Profiles of MT-A70 Genes

The expression of MT-A70 gene family members in different tissues of *P. tangutica* was analyzed. The data obtained from NCBI were used to draw a heat map of specific expression (Figure 7), which showed that *PtMTA1* and *PtMTB2* showed higher expression in flowers and sepals, and the other six genes, except for the *PtMTC1* gene, were more highly expressed in sepals. *PtMTC2* had lower expression in all tissues of *P. tangutica*. In leaves, the relative expression of all seven *PtMT* genes was low (Appendix A).

### 2.9. qRT-PCR

The expression of *PtMT* genes in different tissues was investigated using qPCR (Figure 8). The results showed that the expression of these genes displayed differential expression patterns. All the *PtMT* genes were highly expressed in roots, especially the *PtMTB1* gene which had the highest expression in roots.

## 3. Discussion

m^6^A RNA methylation is the predominant intermediate chemical modification implicated in post-transcriptional gene regulation in eukaryotes. It is essential in governing mRNA processing and metabolism, encompassing translation, degradation, splicing, and transport [33]. Consequently, m^6^A plays a role in the regulation of numerous biological processes. This study utilized a bioinformatics approach to identify and characterize seven members of the MT-A70 gene family in *P. tangutica*, which constitutes a relatively small gene family. Comparable results have been observed in other species, including *Populus alba × Populus glandulosa* (Poplar 84K), which possesses eight MT-A70 genes [34], whereas Litchi (*Litchi chinensis* Sonn.) [35] and *Aegilops tauschii* [36] each harbor two MT-A70 genes. Phylogenetic analyses indicated that the MT-A70 gene family in *P. tangutica* encodes proteins classified into three subclades, with *PtMTBs* representing the largest group, consisting of three genes. This distribution resembles that observed in Arabidopsis. MTA has been thoroughly investigated in Arabidopsis. A backfill mutant strain (mta-ABI3 prom: MTA) that selectively expresses MTA during embryogenesis displays phenotypes including diminished apical dominance, malformed floral organs, augmented epidermal hair meristems, stunted root growth, modified root primordial xylem development, and gravitropic defects [37]. Considering that *P. tangutica* contains two MTA genes that cluster within the same phylogenetic branch as *AtMTA*, it is reasonable to assume they may execute analogous methylation functions.

The ratio of nonsynonymous to synonymous substitutions (*Ka/Ks*) is a crucial metric for evaluating selective pressure on protein-coding genes. This study found that the *Ka/Ks* values for three collinear gene pairs, *PtMTA1* and *PtMTA2* genes, *PtMTB1* and *PtMTB2*, and *PtMTC1* and *PtMTC2* were all under 0.5. This indicates that the *PtMT* gene family has experienced significant purifying selection throughout evolution [38], reflecting the preservation of their structure and function.

Recent studies have emphasized the significance of *PtMT* genes in responses to abiotic stresses, such as drought, salinity, and hormonal regulation [39,40,41]. The expression levels of OsMTA and OsMTB in rice diminished under drought stress. The overexpression of *PtrMTA* in poplar markedly augmented trichome density and promoted root system development, consequently enhancing drought tolerance [42,43]. Furthermore, m^6^A methylation is essential for salt stress tolerance in *Arabidopsis thaliana* [44], as evidenced by the salt-sensitive phenotypes observed in *MTA* and *MTB* mutants. Drought stress similarly prompts the expression of *ClMTB*, a m^6^A methyltransferase in watermelon. The overexpression of *ClMTB* in tobacco plants augments drought tolerance by enhancing reactive oxygen species scavenging activity [45]. The examination of cis-regulatory elements in the promoters of *PtMT* genes indicated that their transcriptional initiation is predominantly governed by light, phytohormones (ABA, GA, MeJA, and SA), environmental stresses, and developmental signals (Figure 4 and Appendix A). Gene Ontology annotation and enrichment analyses revealed that the seven *PtMT* genes predominantly exhibit methyltransferase activity (molecular function), localize to the nucleus (cellular component), and participate in mRNA methylation (biological process) (Figure 3B and Appendix A).

In this study, we analyzed the expression of MT-A70 genes in six tissues of flower, leaf, sepal, early_calyx, mid_calyx, and late_calyx. The results showed that these genes were expressed in all six tissues of *P. tangutica*, but there were differences in the expression profiles with the lowest expression in late_calyx tissues, and that genes clustered on the same branch showed similar expression patterns, suggesting that they may play similar roles in plant growth and development. Previous studies reported that in cotton [46], RNA-seq data showed that *GhMETTL14* and *GhMETTL3* were highly expressed in roots, stems, leafs, torus, petals, stamens, pistils, and calycle tissues, which is consistent with our results. The m^6^A modification level is closely related to the expression of methyltransferase in rice panicles and flag leaves [47], with a more significant difference between different organs than between the same organ at different stages. Therefore, it is necessary to further study the detailed functions of MT-A70 genes in the growth and development of *P. tangutica*.

## 4. Materials and Methods

### 4.1. Identification of PtMTs Gene Family

The reference genome and annotated protein sequences of *P. tangutica* were sourced from the NCBI database (https://www.ncbi.nlm.nih.gov/, accessed on 25 November 2023) [32]. TAIR provided the amino acid sequences of the four MT-A70 proteins from *Arabidopsis thaliana*. Potential MT-A70 members were found by BLASTP analysis (E-value threshold of 1 × 10^−5^) of the *P. tangutica* genome using TBtools software (v2.096) [48]. Conserved structural domains were found using the NCBI-CDD database (http://www.ncbi.nlm.nih.gov/Structure/cdd/wrpsb.cgi/, accessed on 8 December 2023) [49] and the Pfam database (http://pfam.xfam.org/, accessed on 8 December 2023) [50]. Excluded were candidate members lacking the MT-A70 domain or with partial domains; the last PtMT gene sequences were assigned sequentially according to their chromosomal sites.

### 4.2. Physicochemical Properties Analysis

*PtMT* proteins’ physicochemical characteristics, including grand average hydropathicity (GRAVY), aliphatic index, molecular weight (MW), instability index, and isoelectric point (pI), were investigated using the ExPASy website (http://web.expasy.org/protparam/, accessed on 12 February 2024). The WoLFPSORT website (https://wolfpsort.hgc.jp/, accessed on 12 February 2024) was used to predict subcellular localization [51].

### 4.3. Gene Structure, Conserved Motifs, and 3D Protein Analysis

GSDS 2.0 (http://gsds.cbi.pku.edu.cn/, accessed on 15 February 2024) was given CDS sequences and their related genomic sequences for the examination of exon–intron structures and visualization [52]. Using MEME (https://meme-suite.org/meme/, accessed on 15 February 2024) [53], conserved motifs in candidate protein sequences were predicted with a maximum of 10 motifs and default settings. Using GOR IV (https://npsa-prabi.ibcp.fr/cgi-bin/npsa_automat.pl?page=/NPSA/npsa_gor4.html/, accessed on 17 February 2024) [54], protein secondary structure was predicted; three-dimensional structural models were produced using the SWISS-MODEL online tool (https://swissmodel.expasy.org/, accessed on 17 February 2024) [55].

### 4.4. Phylogenetic Analysis

A multiple sequence alignment of m6A-related proteins from *Solanum lycopersicum*, *Capsicum annuum*, *Solanum tuberosum*, *Arabidopsis thaliana*, *Nicotiana tabacum*, *Anisodus acutangulus*, *Mandragora chinghaiensis*, *Datura stramonium*, *Anisodus tanguticus*, and *Atropa belladonna* was conducted utilizing ClustalW (https://www.genome.jp/tools-bin/clustalw, accessed on 21 February 2024) [56]. Using the neighbor-joining (NJ) technique in MEGA 11.0 software [57], a phylogenetic tree was produced with the bootstrap parameter set to 1000 and visualized using iTOL (https://itol.embl.de/, accessed on 22 February 2024) [58].

### 4.5. Enrichment Analysis Using Gene Ontology (GO) and Cis-Regulatory Elements

Promoter sequences of *PtMT* genes (2000 bp upstream of the translation initiation codon ‘ATG’) were extracted and analyzed utilizing the PlantCARE database (http://bioinformatics.psb.ugent.be/webtools/plantcare/html/, accessed on 6 March 2024) [59]. The distribution patterns of cis-regulatory elements were visualized with TBtools software [48]. TBtools was used to visualize the *PtMT* genes chromosomal location data obtained from *P. tangutica* genome annotation files [32]. *PtMT* genes were BLASTed against the uniprot_sprot.fasta file retrieved from the STRING database (https://cn.string-db.org/, accessed on 6 March 2024) for GO study [60] and TBtools was used to perform GO annotation and enrichment analysis.

### 4.6. Gene Expression Pattern and Collinearity Analysis

Sourced from the NCBI Sequence Read Archive (SRA) database, eighteen RNA-seq samples were acquired under BioProject accession number PRJNA791792 (https://www.ncbi.nlm.nih.gov/sra/, accessed on 8 March 2024). Using TBtools software [48], the samples acquired from six different *P. tangutica* tissues (Appendix A) were processed for sequence transformation; Galaxy (https://usegalaxy.org/, accessed on 6 March 2024) was used for data analysis [61]. Using ChiPlot (https://www.chiplot.online/, accessed on 10 March 2024), a heatmap showing gene expression patterns was generated. While Advanced Circos of TBtools [62] was used to show the collinear connections inside the *PtMT* gene family, TBtools software was used to extract the *PtMT* gene location data and collinear gene pairs from the *P. tangutica* genome annotation file. The Simple *Ka/Ks* Calculator function in TBtools was used to calculate the non-synonymous (*Ka*) and synonymous (*Ks*) substitution rates for collinear gene pairs [47].

### 4.7. Quantitative Real-Time Reverse Transcription-PCR (qRT-PCR)

Total RNA was extracted from the stems, leaves, and roots of aseptic *P. tangutica* seedlings that were three months old using the SteadyPure Plant RNA Extraction Kit (Accurate Biology, Lanzhou, China). Using the *Evo M-MLV* RT for PCR Kit (Accurate Biology, China), reverse transcription produced cDNA. Adhering to standard procedures, the SYBR Green Premix *Pro Taq* HS qPCR Kit (Accurate Biology, China) was used to run qRT-PCR tests on *PtMT* genes. The reference gene [40] was *PGK*. Appendix A lists primer sequences for all genes; relative gene expression levels were calculated using the 2^–ΔΔCt^ approach [63].

## 5. Conclusions

In this study, 7 *PtMT* genes were identified with the help of bioinformatics tools and the *P. tangutica* genome primarily, they belong to the subfamilies MTA, MTB, and MTC. Additionally, their physicochemical properties, including their phylogenetic relationship, gene and protein structure, chromosomal arrangement, and cis-regulatory elements, were analyzed as part of a comprehensive survey. Expression analysis revealed that different tissues of *PtMT* genes were differentially expressed. Our results will supply some insight into the characteristics of *P. tangutica* MT-A70 genes and their probable epigenetic regulation mechanism in *P. tangutica.*

## Figures and Tables

**Figure 1 ijms-26-03593-f001:**
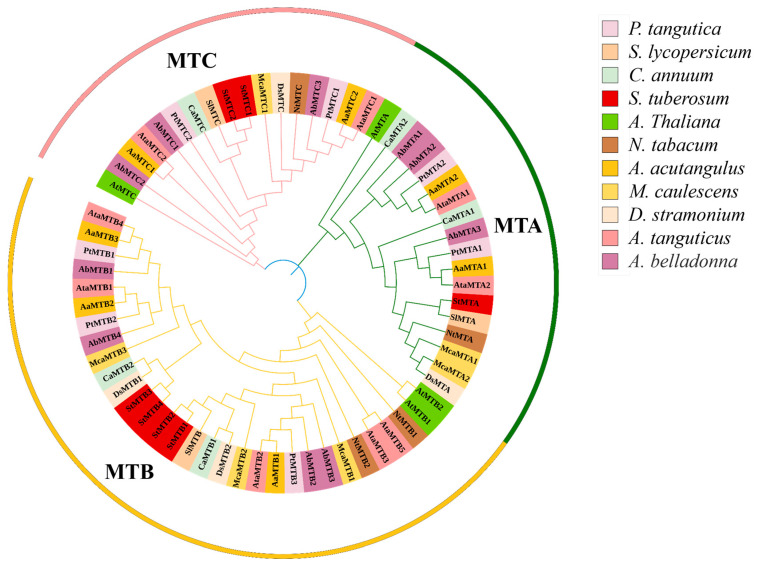
Phylogenetic analysis of MT-70 homologous proteins from *Solanum lycopersicum*, *Capsicum annuum*, *Solanum tuberosum*, *Arabidopsis thaliana*, *Nicotiana tabacum*, *Anisodus acutangulus*, *Mandragora chinghaiensis*, *Datura stramonium*, *Anisodus tanguticus* and *Atropa belladonna*. The phylogenetic trees were constructed using MEGA 11.0 by the neighbor-joining (NJ) method with 1000 bootstrap replicates. The groups of m^6^A pathway genes from *Przewalskia tangutica* are shown in different colors.

**Figure 2 ijms-26-03593-f002:**
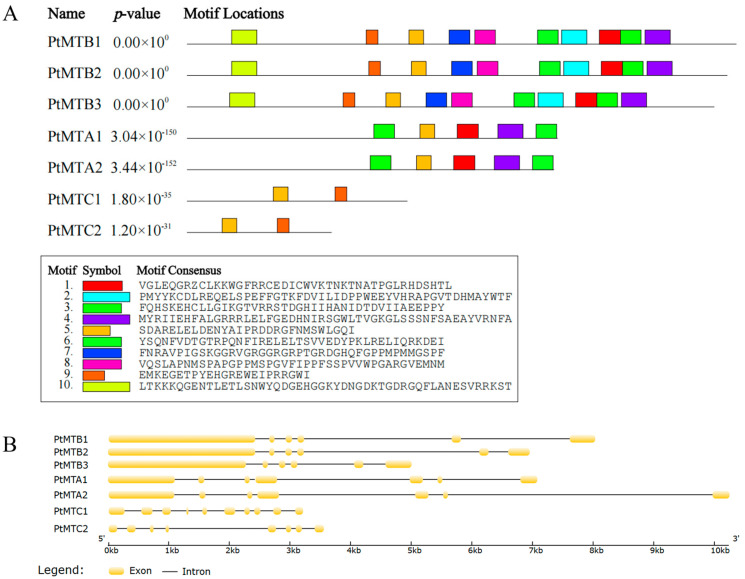
The gene structure and conserved motifs of *PtMT* genes. (**A**) Exons and intron phases are shown. (**B**) The distribution of conserved motifs.

**Figure 3 ijms-26-03593-f003:**
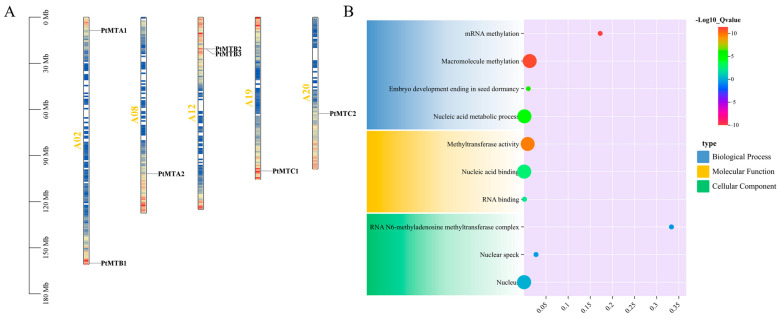
Chromosomal location and GO enrichment analysis of *PtMT* genes. (**A**) Chromosomal distribution. (**B**) GO enrichment analysis. The color of circles are colored according to the −Log10_Qvalue. The size of the circles is determined by the number of annotated genes.

**Figure 4 ijms-26-03593-f004:**
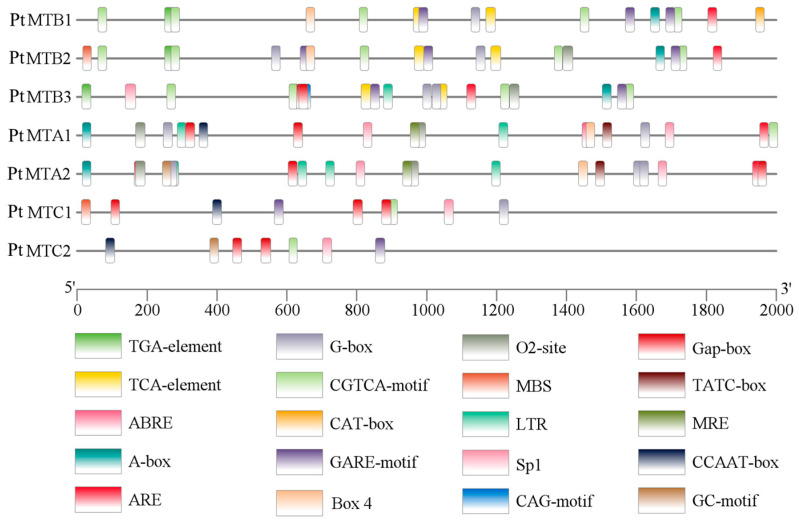
The cis-regulatory elements in the promoter of *PtMT* genes. The cis-regulatory elements were represented as rectangles in different colors.

**Figure 5 ijms-26-03593-f005:**
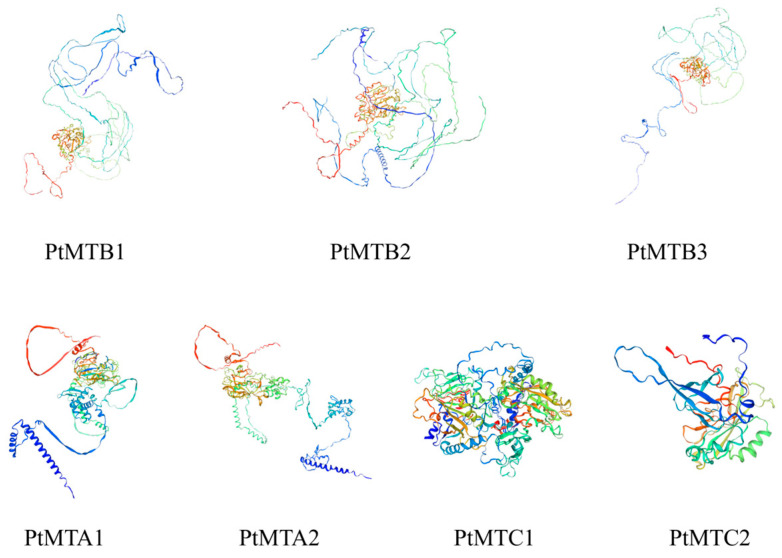
Prediction of the three-dimensional domain of PtMT proteins, the structure of the proteins is rainbow-colored, with the N-terminus shaded blue and C-terminus shaded red.

**Figure 6 ijms-26-03593-f006:**
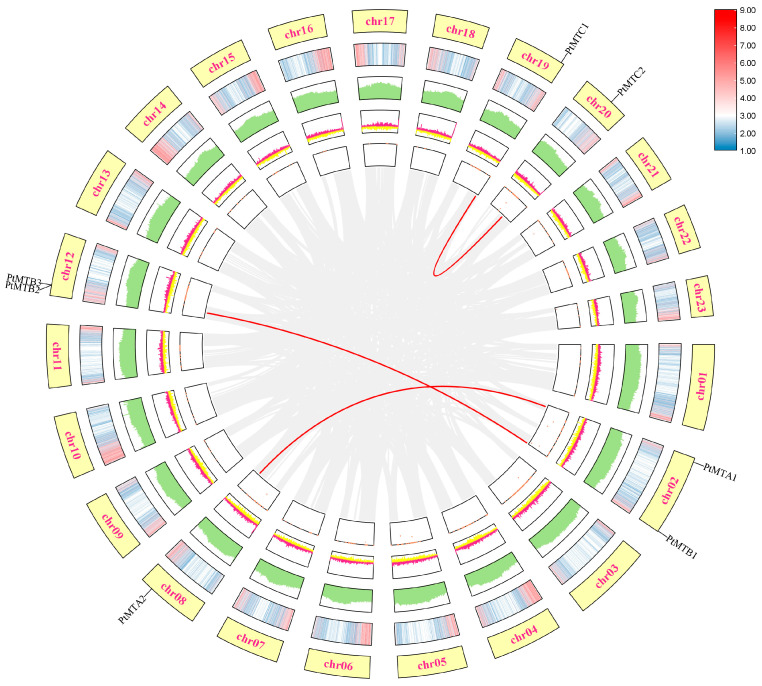
Collinearity analysis of *PtMT* gene family members, the red line indicates that there are collinear relationships between *PtMT* gene family members.

**Figure 7 ijms-26-03593-f007:**
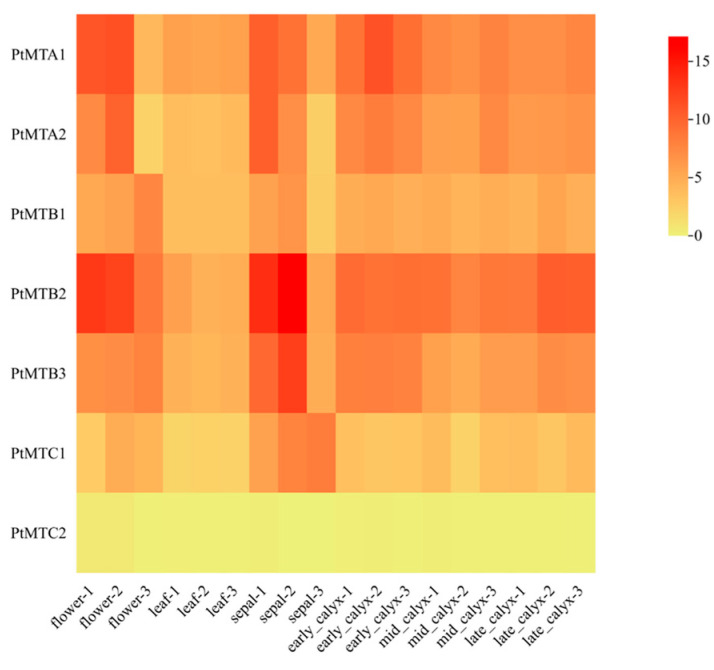
Expression profiles of *PtMT* genes under six different tissues.

**Figure 8 ijms-26-03593-f008:**
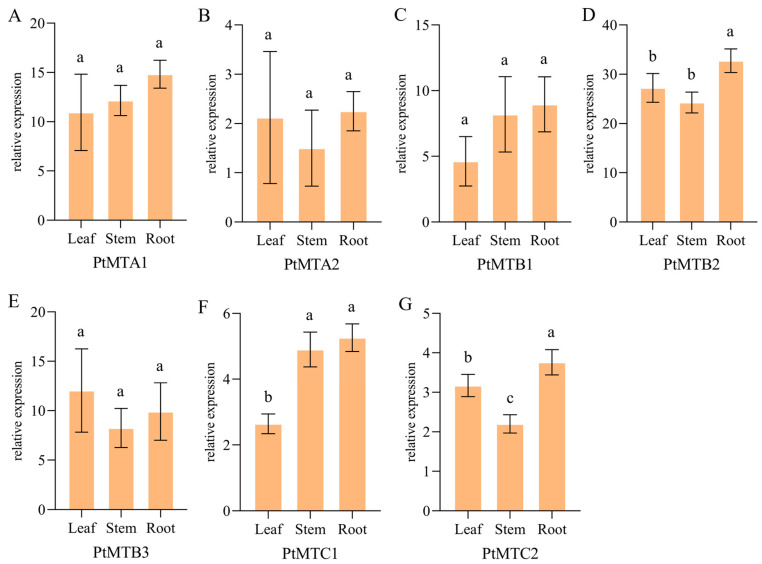
qRT-PCR validation of 7 *PtMT* genes in tissues. Figure (**A**–**G**) showed the expression level of 7 genes in differential tissues. One-way ANOVA was calculated using IBM SPSS 27 software. The a, b, and c indicated whether the difference was significant. The same letter marked in the same gene among different tissues indicated no significant difference, and different letters indicated significant differences.

**Table 1 ijms-26-03593-t001:** Sequences feature of *PtMT* genes.

Genes	Length (bp)	Length (aa)	MW (kDa)	pI	Instability Index	Aliphatic Index	Grand Average of Hydropathicity (GRAVY)	Subcellular Localization Predicted
*PtMTA1*	2217	738	81.83	6.72	42.05	78.86	−0.461	chloroplast
*PtMTA2*	2196	731	80.63	6.47	41.33	79.48	−0.391	chloroplast
*PtMTB1*	3285	1094	122.96	6.16	57.55	48.39	−1.155	nucleus
*PtMTB2*	3231	1076	120.82	6.14	58.2	48.84	−1.163	nucleus
*PtMTB3*	3153	1050	117.53	6.77	56.36	50.99	−1.102	nucleus
*PtMTC1*	1320	439	50.13	7.1	49.4	81.3	−0.389	nucleus
*PtMTC2*	867	288	32.54	5.12	45.79	83.26	−0.239	cytoplasm

## Data Availability

Data are contained within the article and Appendix A.

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
