# Peer review of "Genome-Wide Identification and Expression Analysis of m6A Methyltransferase Family in Przewalskia tangutica Maxim."

_ijms, 2025, doi:10.3390/ijms26083593_

Round 1
Reviewer 1 Report
Comments and Suggestions for Authors
In the submitted study the authors formed a genome-wide characterization of PtMT genes (PtMT1-7) and a comprehensive analysis of their phylogenetic relationship, gene and protein structures, chromosomal arrangement, cis-regulatory elements, expression and physicochemical properties. The manuscript is well done, but only sustained by bioinformatics, thus appearing descriptive in all parts and quite limited for the IJMS.
My primary concern is about the suitability of this manuscript to the IJMS: most data are from bioinformatics; expression analysis in 3 different tissues only was carried out to validate bioinformatics data. What about proteins and their biological activity? What about the molecular mechanism and the biological relevance of this comparative study?
Figure1 legend: add the accession number of the used sequences
195-195 “All the PtMT genes were highly expressed in roots, especially, the 196 PtMTB1 genes which had the highest expression in roots.” This sentence is not consistent to the provided statistics in the graphs (data are statistically significant for PtMTB2, PtMTC1 and PtMTC2 only)
English style requires slight revision
The name of species always needs Italic font.
Author Response
Thank you very much for your time and effort in reviewing our manuscript. Your comments are highly valuable and have helped us to significantly improve our work.
1.Relevance of References
We conducted a thorough review of all references cited in the manuscript. For instance, We added eight relevant new references in the Materials and Methods section. these newly added references provide more up-to-date and relevant data, which will help to strengthen the scientific basis of our research.
2. Highlighting Revisions
To make it easy for you and the reviewers to track the changes, we used the “Track Changes” feature in Microsoft Word. All revisions, including text additions, deletions, and modifications, are clearly marked. This way, you can quickly identify the modifications we have made to the manuscript and assess their impact on the overall content.
Reviewer 2 Report
Comments and Suggestions for Authors
The researchers conducted an interesting study that contributes to the genetic improvement of the species. However, there are some aspects that could improve the quality of the manuscript, which are described below:
The title is relevant and appropriate to the nature of the study.
While the abstract contains the basic elements, it needs to emphasize the biological, genetic, and agronomic impact of the results.
Review the keywords to ensure greater citation of the article.
In the introduction, background studies should be contextualized within the national and regional context. Some referenced studies could be more recent.
In the results, improve the presentation of tables and figures.
In the discussion section, emphasize not only the biological but also the genetic and agronomic impact of the results.
Improve the conclusions, as they should demonstrate the real impact of the results.
Some references could be updated.

Overall the manuscript is understandable, however, there are some minor errors noted in the document.
Author Response

(The authors gave the same response as above.)
